

# A Generalized Simulation Capability for Rotating Beam Scatterometers

Zhen Li[1], Ad Stoffelen[1], Anton Verhoef[1]

[1]R&D Satellite Observation, Royal Netherlands Meteorological Institute, de Bilt, 3731 GA, Netherlands

*Correspondence to*: Zhen Li (li@knmi.nl)

**Abstract.** Rotating-beam wind scatterometers exist in two types: rotating fan-beam and rotating pencil-beam. In our study, a generic simulation frame is established and verified to assess the wind retrieval skill of the three different scatterometers: SCAT on CFOSAT, WindRad on FY-3E and SeaWinds on QuikScat. Besides the comparison of the so-called 1st rank-solution retrieval skill of the input wind field, other Figure of Merits (FoMs) are applied to statistically characterize the associated wind retrieval performance from three aspects: wind vector root mean square error, ambiguity susceptibility, and wind biases. The evaluation shows that, overall, the wind retrieval quality of the three instruments can be ranked from high to low as WindRad, SCAT, and SeaWinds, where the wind retrieval quality strongly depends on the Wind Vector Cell (WVC) location across the swath. Usually, the higher the number of views, the better the wind retrieval, but the effect of increasing the number of views reaches saturation, considering the fact that the wind retrieval quality at the nadir and sweet swath parts stays relatively similar for SCAT and WindRad. On the other hand, the wind retrieval performance in the outer swath of WindRad is improved substantially as compared to SCAT due to the increased number of views. The results may be generally explained by the different incidence angle ranges of SCAT and WindRad, mainly affecting azimuth diversity around nadir and number of views in the outer swath. This simulation frame can be used for optimizing the Bayesian wind retrieval algorithm, in particular to avoid biases around nadir, but also to investigate resolution and accuracy through incorporating and analysing the spatial response functions of the simulated Level-1B data for each WVC.

## 1 Introduction

The wind scatterometer has been proven to be a powerful instrument for global sea surface wind measurement. The wind retrievals have a wide variety of applications, including now-casting, and assimilation in numerical weather prediction models, as well as oceanography, climate research, and off-shore energy applications ( Offiler, 1984; Naderi et al., 1991; Stoffelen and Anderson, 1997; Portabella, 2002; Bajo et al., 2017). The wind retrieval is achieved by inverting a set of radar cross-section measurements ($\sigma°$) at different geometries (incidence and/or azimuth look angles) over a Wind Vector Cell



(WVC) through a Geophysical Model Function (GMF) to extract the wind. The more diversity in the geometry, the better wind retrieval will be achieved (Portabella, 2002).

Currently, there are two types of scatterometer in orbit: multiple fixed fan-beam and rotating pencil-beam instruments. The first wind scatterometer in space was the SEASAT-A Scatterometer System (SASS) on SEASAT-A launched in June 1978 by NASA with four fixed fan beams and dual co-polarization (VV and HH) Ku-band (13.2 GHz) emitting and receiving antennas, which failed in October 1978 (Offiler, 1984). The term 'views' in this paper means measurements of the surface $\sigma^o$ at different azimuth angle and/or incidence angle and/or polarizations, and each surface $\sigma^o$ measurement is aggregated from the samples with the same polarization, similar azimuth and incidence angle. The geometric diversity of the views is able to improve the wind retrieval accuracy. 'Views' is different from the term 'looks' in radar which is defined as the equivalent number of independent samples in a particular $\sigma^o$ measurement and specifies the measurement variance (Ulaby and Long, 2013). This scatterometer had two views only per Wind Vector Cell (WVC), a VV view and an HH view, which turned out insufficient to well resolve the wind direction unambiguously. The ERS-1 and -2 satellites carried a scatterometer onboard as of 1991 three fixed fan beams and vertical co-polarization (VV) at C-band frequency (5.4 GHz), with all beams pointing to the right-hand side of the satellite. After ERS-1/2, the NASA Scatterometer (NSCAT) was launched in 1996 on the Japanese Advanced Earth Observing Satellite (ADEOS-I). It had six fan beams with VV capability on the fore and after beams, and both VV and horizontal (HH) co-polarization on the mid beams (Naderi et al., 1991). The European Space Agency (ESA) developed the Advanced Scatterometer (ASCAT) on the Metop satellite series, which has six C-band VV fan beams, each 3 pointing to the left and right of the swath resp., and it started to provide data in 2006 (Gelsthorpe et al., 2000). The ERS-1/2, NSCAT and ASCAT instruments all use three independent views per WVC, leading to a reduced wind direction ambiguity as compared to SASS, by well sampling the main second harmonic wind direction dependency of the Geophysical Model Function (GMF) (Stoffelen and Anderson, 1997; Stoffelen and Portabella, 2006). SeaWinds, the first rotating pencil-beam scatterometer was developed by NASA and launched on QuikSCAT (1999), on the Japanese satellite ADEOS-2 (2003) and flew as RapidScat on the International Space Station in 2014. It has two Ku-band rotating pencil-beams measuring VV and HH, respectively, at two fixed incidence angles (Hoffman and Leidner, 2005). All current and prior rotating pencil-beam scatterometers are similar in design concept to SeaWinds and differ primarily in the used incidence angles. The OSCAT scatterometer on Oceansat-2 is a Ku-band rotating pencil-beam instrument similar to SeaWinds and developed by the Indian Space Research Organization (ISRO). It was launched in 2009 and failed in 2014 (Singh et al., 2012). After that, ISRO launched SCATSat-1 in 2016 as an OceanSat-2 replacement mission with the same scatterometer design and OceanScat-3 will be launched in 2018. China launched its first Ku-band rotating pencil-beam scatterometer on board HY-2A in 2011 and operating until present (Jiang et al., 2012). SeaWinds-class rotating pencil-beam scatterometers are able to obtain four independent views per WVC in the inner swath, but only two independent views per WVC in the outer swath, where only vertically polarized views are available. This will impose similar ambiguity problems as in the SASS design.

A new type of scatterometer – the Rotating Fan-beam Scatterometer (RFSCAT) in Ku-band was proposed in 2000 (Lin et al., 2000b). It combines the features from fixed fan-beam and rotating pencil-beam scatterometers, which provide large



swath coverage and increase the diversity in the observation geometry. The scatterometer (referred to as SCAT from now on) onboard CFOSAT (China-France Oceanography SATellite) and WindRad (Chinese Wind Radar on FY-3E) belong to this type of scatterometer and are planned to be launched in 2018. These represent a rotating fan-beam instrument with Ku-band only (SCAT), a rotating fan-beam instrument with both Ku and C-band (WindRad), and a rotating pencil-beam instrument with Ku-band only (SeaWinds).

The aim of our study is to build a generic simulation system and construct an evaluation frame, particularly fit for the above rotating-beam scatterometers, including Ku-band and C-band. The simulation system includes the complete simulation of satellite orbital movement, Level-1B (L1B) data generation, Level-2A (L2A) data generation and Level-2B (L2B) wind retrieval. The three different rotating-beam scatterometers are expected to perform differently, due to their varying observation geometry and non-linear wind retrieval characteristics, e.g., wind direction ambiguity. The wind retrieval results are carefully evaluated and compared. The advantages and disadvantages are analyzed such that they can be used as design reference.

## 2 Simulation method

### 2.1 CFOSAT, WindRad and SeaWinds characteristics

The RFSCAT characteristics have been studied and assessed by Lin et al. (Lin et al., 2000a, 2002). The slowly rotating fan-beam sweeps over the swath and the different views overlap in each WVC, which leads to multiple views in a given WVC (Figure 1). Contrary to the fixed fan-beam and rotating pencil-beam instruments, the number of views in a WVC depends on its location and varies across the swath as a function of the rotating speed. The scanning geometry results in a smaller number of views and less azimuth diversity in the outer and the nadir parts of the swath, which lead to a degraded wind retrieval performance. In contrast, the other region of the swath (named as sweet swath) has a better wind retrieval performance than the outer and nadir swath.

SCAT and WindRad are both rotating fan-beam designs, but they have somewhat different characteristics. They both follow the RFSCAT principles, but SCAT has two fan beams operating in Ku-band with VV and HH respectively, whereas WindRad has four fan beams. Two of these beams are operating in Ku-band at VV and HH respectively while the other two are operating VV or HH in C-band. All the antennas transmit and receive pulses in turns (see the illustrations in Figure 1). The main parameters for simulating SCAT and WindRad are listed in Table 1 and Table 2.

Rotating pencil-beam scatterometers have been flying on several satellites as described in the Introduction. SeaWinds is taken as representative for the rotating pencil-beam design in our study. It has one dish antenna of about 1 m diameter with a VV and HH beam conically scanning at a speed of 18 rpm, which is much faster than the rotating fan-beam (Figure 2). The VV beam has a higher incidence angle than the HH beam, resulting in a wider VV swath. There are four integrated views produced at all WVCs; for those located in the inner swath by segregating both VV and HH and fore and aft views. The four views in the outer swath that are used in the retrieval are all VV views, also divided into resp. fore and aft views, but each split in two azimuth groups. The main parameters of the SeaWinds instrument are listed in Table 3.

30



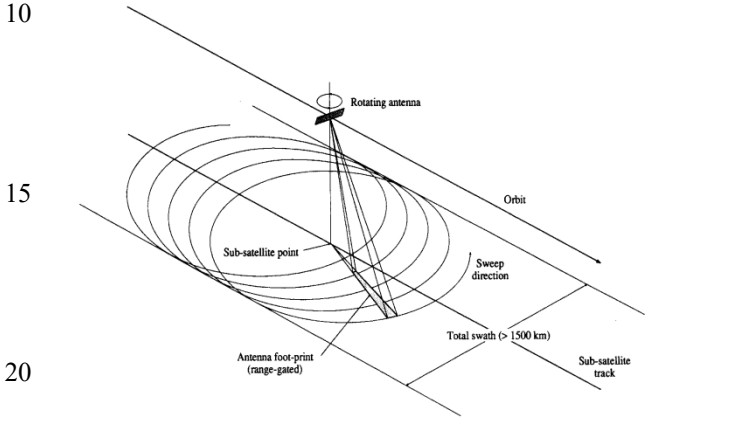

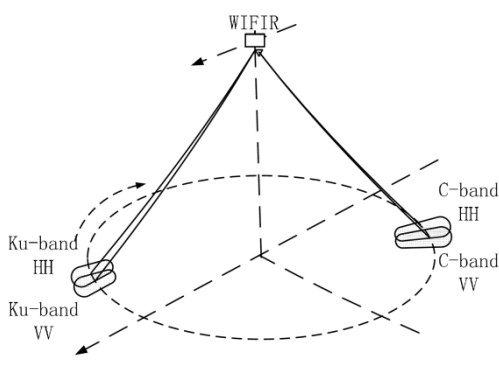

(a)                     (b)

**Figure 1: Rotating fan-beam scatterometer. (a) SCAT (Lin et al., 2000a); (b) WindRad (Dou et al., 2014).**

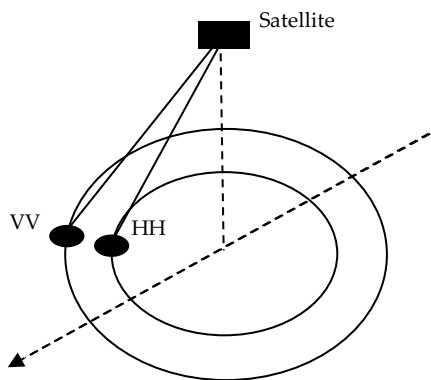

**Figure 2: Rotating pencil-beam scatterometer.**



**Table 1. Main parameters of CFOSAT SCAT**

| Parameters | Value |
| --- | --- |
| Orbit height | 514 km |
| Swath | 1000 km |
| Footprint | 280 km |
| Satellite speed | 7.1 km/s |
| Antenna rotating speed | 3.5 rpm |
| Polarization | VV and HH alternating |
| Incidence angle range | 25 – 48 deg |
| Antenna pointing angle | 40 deg |
| Peak transmit power | 120 W |
| WVC resolution | 25 km |
| Antenna bandwidth | 13.256 GHz (Ku-band) |
| Duration of transmit pulse | 1.3 ms |
| Duration of receiving pulse | 2.7 ms |
| Pulse Repetition Frequency (PRF) | 75 Hz |
| Two-way -3dB beam width (azimuth) | 1.28 deg |
| Peak antenna gain | 30 dB |

**Table 2. Main parameters of FY-3E WindRad**

| Parameters | Value | |
| --- | --- | --- |
| | Ku-band | C-band |
| Orbit height | 836 km | |
| Swath | 1400 km | |
| Footprint | 200 km | |
| Satellite speed | 7.4 km/s | |
| Antenna rotating speed | 3.0 rpm | |
| Polarization | VV and HH alternating | |
| Incidence angle range | 34.7 – 44.5 deg | |
| Antenna pointing angle | 34.8 deg | |
| WVC resolution | 25 km | |
| Peak transmit power | 120 W | 100 W |
| Antenna bandwidth | 13.256 GHz | 5.4 GHz |
| Duration of transmit pulse | 1.8 ms | 1.7 ms |
| Duration of receiving pulse | 1.25 ms | 1 ms |
| Pulse Repetition Frequency (PRF) | 208 Hz | 104 Hz |



| Two-way -3dB azimuth beam width | 1.3 deg | 0.52 deg |
| Peak antenna gain | 37 dB | 32 dB |

**Table 3. Main parameters of QuikScat SeaWinds**

| Parameters | Value (inner and outer beam) |
|---|---|
| Orbit height | 800 km |
| Swath | 1800 km |
| Footprint | 36 km |
| Satellite speed | 7.0 km/s |
| Antenna rotating speed | 18 rpm |
| Polarization | VV and HH |
| Incidence angle range | 51.8 deg and 46.7 deg |
| Antenna pointing angle | 44.9 deg and 38.9 deg |
| Peak transmit power | 120 W |
| WVC resolution | 25 km / 12.5 km |
| Antenna bandwidth | 13.256 GHz (Ku-band) |
| Duration of transmit pulse | 1.5 ms |
| Duration of receiving pulse | 2.1 ms |
| Pusle Repetition Frequency (PRF) | 96 Hz |
| Two-way -3dB beam width (azimuth) | 1.8 deg |
| Peak antenna gain | 38 dB |

**1.2 Simulation procedure**

10   The simulation is designed to be generic and able to adapt to all of the current rotating-beam wind scatterometers, i.e., both pencil beam and fan beam. It consists of four components: (1) generate satellite state vectors by the orbit propagator SGP4 (Simplified perturbations models) (Hoots and Roehrich, 1980); (2) simulate L1B data; (3) assign the L1B data onto the proper WVCs; (4) aggregate L1B data in one WVC into views (L2A data). The work flow charts are shown in Figure 3 and Figure 4. We use ECMWF model wind as input wind field to initialize the L1B simulation, which provides a spatially

15   smooth ocean wind truth. To represent the sampling of local wind variability (turbulence), geophysical noise is added by disturbing the input wind components u and v assigned on each slice by injecting Gaussian distributed noise. Together with the instrument configurations and satellite state vectors, the observation geometries on slice level are calculated. The



instrument noise Kpc (Long et al., 2004) is estimated by $K_{pc}^2 = A + \frac{B}{SNR} + \frac{C}{SNR^2}$. However, the coefficients A, B, and C need onboard processing details, which are not the same nor available for all scatterometers. In order to make the simulator generic, A, B, and C for each slice are calculated by $A = \frac{1}{B_s \times t_d}$, $B = \frac{2}{B_s \times t_r}$, $C = \frac{1}{B_s \times t_r}$, where $B_s$ is the bandwidth for each individual slice, $t_d$ is the transmit duration time, $t_r$ is the receiving time. The distribution of $B_s$ on each slice in one pulse is assigned according to the antenna gain pattern of the pulse.

An example of the simulated satellite orbit together with the location of the slices is given in Figure 5. $\sigma°$ is derived using the NSCAT-4 GMF for Ku-band and the CMOD5n GMF for C-band and the corresponding beam geometries. Subsequently, the L1B data are obtained after adding the instrument noise on the 'true' $\sigma°$. The instrument noise is added by multiplying a Gaussian random number in this way: $\sigma°_{noise} = \sigma° \times (1 + K_{pc} \times Gaussian\_random\_nr)$. The L1B data are assigned to the proper WVCs (Dunbar et al., 2001) and then aggregated into views. A view is a group of slices with similar azimuth angle and the same polarization in one WVC, the properties (i.e. incidence angle, azimuth angle, latitude, longitude, etc.) on the corresponding slices are also aggregated to represent the view (Li et al., 2017). We note that the simulation does currently not include rain effect.

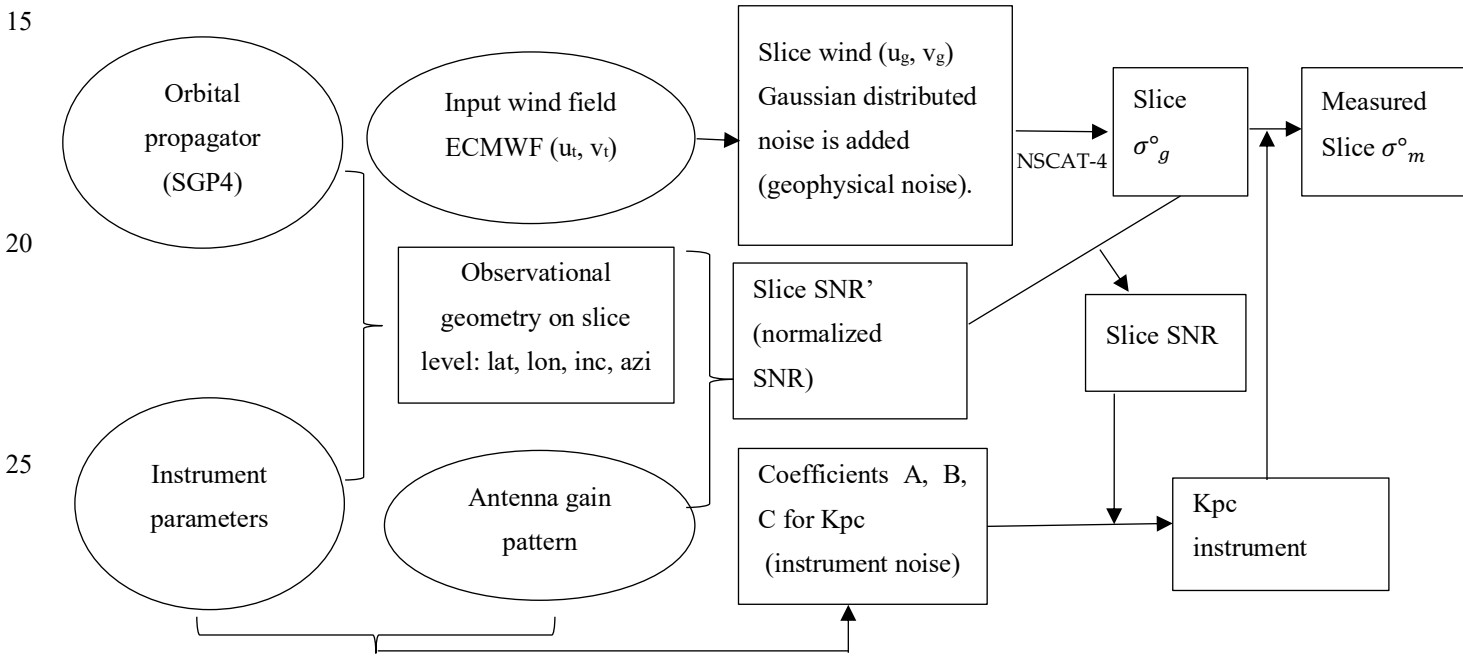

**Figure 3: The workflow for generating L1B simulation data.**





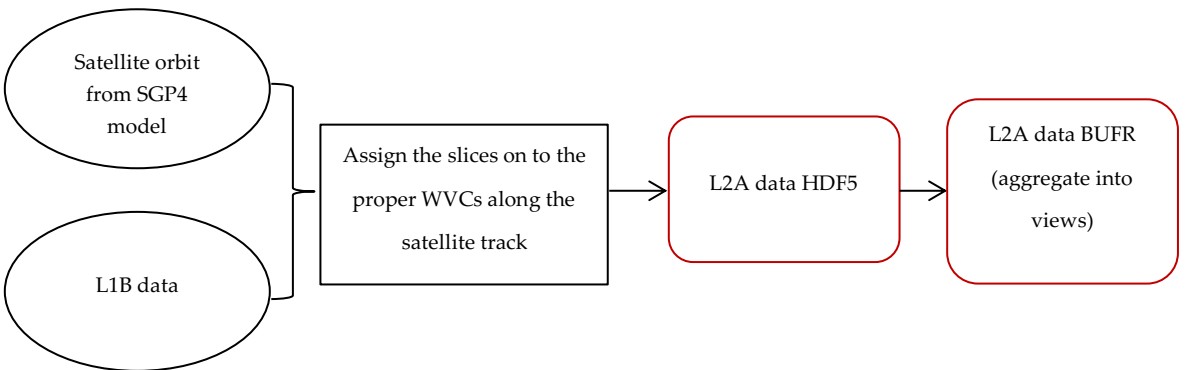

**Figure 4: The workflow to assign L1B data to the proper WVCs and aggregate into views.**

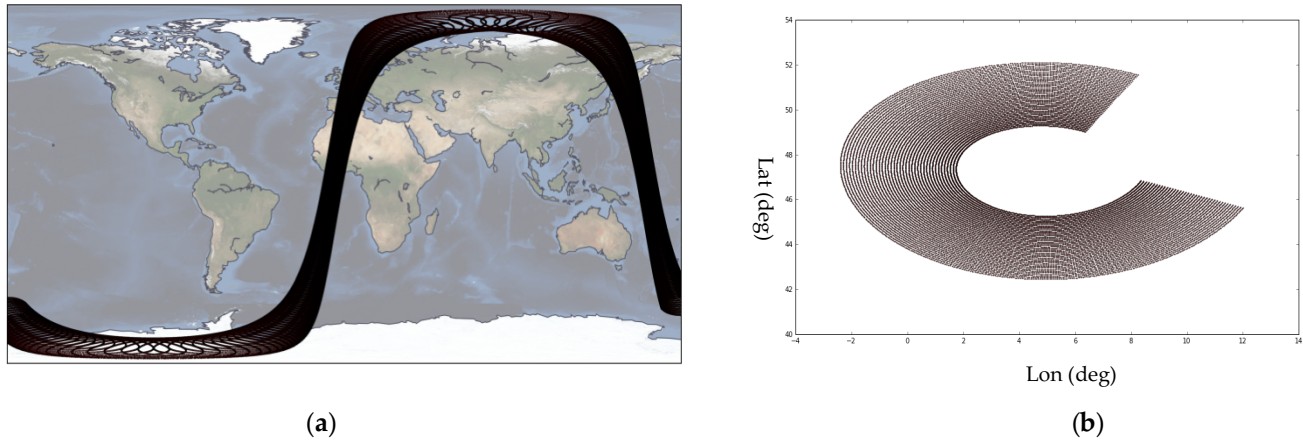

(a)                                                              (b)

**Figure 5: (a) One simulated satellite orbit for CFOSAT starting from 11-12-2011 with the circular motion of the slice located at the**
10   **end of each pulse; (b) the zoomed in location of all slices on the earth.**

**2.3 Wind field retrieval principle**

The Maximum Likelihood Estimator (MLE) is the most classic algorithm for wind retrieval. It has been applied in
many wind retrieval studies (Chi and Li, 1988; JPL, 2001; Pierson, 1989; Portabella and Stoffelen, 2002). We adopted it and
applied it in our wind retrievals. The MLE can be expressed as (JPL, 2001):

$$MLE = \frac{1}{N}\sum_{i=1}^{N}\left(\frac{\sigma^{\circ}{}_{mi} - \sigma^{\circ}{}_{si}}{Kp(\sigma^{\circ}{}_{xi})}\right)^2 \tag{1}$$

15   where N is the number of views, and $\sigma^{\circ}{}_{xi}$ is either $\sigma^{\circ}{}_{mi}$ (measured $\sigma^{\circ}$) or $\sigma^{\circ}{}_{si}$ (trial simulated $\sigma^{\circ}$). $Kp(\sigma^{\circ}{}_{xi})$ is the expected
Gaussian observation noise with the form of $Kp \times \sigma^{\circ}{}_{xi}$. The wind inversion procedure takes L2A data and searches for the





$\sigma°_{si}$ with minimum MLE by varying trial wind speeds and directions. The $\sigma°_{si}$ with the minimum MLE is known as the first rank solution. However, the first solution is often not the best solution because the wind retrieval results usually consist of a set of ambiguous solutions due to the combination of measurement geometry, the harmonic modulation of the GMF (non-linear GMF), noise, etc. After the wind retrieval step, one of the ambiguous solutions is selected by the Two-Dimensional

Variational Ambiguity Removal (2DVAR) (Vogelzang, 2013) after minimizing a total cost function that combines both observational and NWP background contributions. The retrieved wind field can be compared with the input wind field to assess the wind retrieval quality.

### 2.3 Simulation assessment

### 2.3.1 SCAT, WindRad, and SeaWinds view number comparison

The most important differences between SCAT, WindRad, and SeaWinds are the shape of the antenna and the number of antennas, directly leading to a different distribution of the number of views across the swath. SeaWinds as rotating pencil-beam instrument has 4 views in each WVC across the swath, where the fore and aft views in the outer swath are each split in two views. The number of slices in each view varies across the swath though. For rotating fan-beam instruments, the view number varies across the swath with the feature of less views in the outer and nadir swath, and more views in the parts of the

swath in between (Figure 6). It can be observed that both SCAT and WindRad contain more views than SeaWinds for all WVCs, with a saddle shape in the view count. Moreover, the number of views of WindRad is about twice the number of views of SCAT.

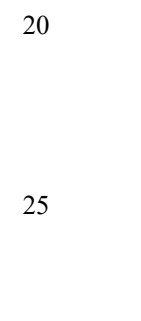

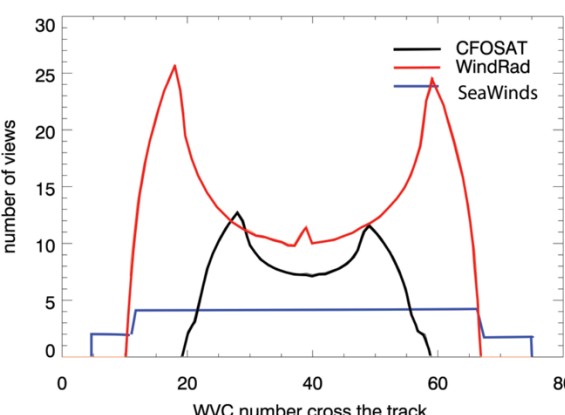

**Figure 6. Averaged number of views at the WVCs across the swath.**



### 2.3.2 Instrument noise

The instrument noise (Kpc) of the simulator for SCAT, WindRad and SeaWinds are estimated at various wind speed (4 m/s, 9 m/s, and 16 m/s) on slice level and WVC level. The Kpc for each view is aggregated by weighting the Kpc of the slices in
this view and the Kpc on WVC level is derived by averaging the Kpc for all the views in the corresponding WVC.

Figure 7 (a) shows the slice Kpc of SCAT as a function of incidence angle. The slices with high wind speed and high incidence angle contain high Kpc and Kpc for VV polarization overall is lower than for HH, except for the slices with incidence angle lower than 30.25° (indicated by the dashed line in Figure 7a). The Kpc in a WVC for SCAT (Figure 7 (c)) is much lower than the Kpc on slice level, as expected due to the aggregation of the slices in a WVC. The outer swath contains
relatively high instrument noise as compared to sweet and nadir swath. Low wind speed leads to a higher Kpc. On the WVC level, the instrument noise is lower than 20% except for low wind speed.

WindRad has two frequencies at Ku and C band. As illustrated in Figure 8 (a) and (b), the VV Kpc is lower than the HH Kpc for Ku and C band and the C-band Kpc is much lower than the Ku-band Kpc. On the WVC level (Figure 8 (c)), it shows a similar pattern to SCAT and generally the instrument noise is lower than 10% if the outer swath and low wind
speeds are excluded.

The SeaWinds Kpc on slice level (Figure 9 (a) (b)) is more constant at wind speed of 9m/s and 16m/s, while it is increasing along with the incidence angle at low wind speed 4m/s. On WVC level, the Kpc is lower than 20% except for wind speeds below 4m/s. We note that a random error of 20% at 4 m/s is still acceptable in terms of absolute random wind error after wind retrieval.

In general, low wind speeds cause high instrument noise, as expected, and the instrument noise on WVC level is less than 20% for SCAT, less than 10% for WindRad and less than 20% for SeaWinds, when the outer swath and low wind speeds are excluded. All scatterometers above have a pattern of higher Kpc at the outer swath as compared to the other parts of the swath.

### 2.3.3 1st rank wind retrieval and 2DVAR performances

As already known, the first rank solution is not always the best solution, but the more often the first rank solutions are chosen to be the best solution, the lower the ambiguity in the inverted instrument wind solutions. The quality of the 1st rank solution thus reflects the ambiguity in the wind measurement system. So, it is chosen for the comparison of the wind retrieval performance on SCAT, WindRad, and SeaWinds. The difference between 1st rank solution and 2DVAR
performance provides insight in the effects of the wind direction ambiguities on the final selected wind field, which may depend on measurement geometry. 2DVAR with MSS (Multiple Solution Scheme) (Vogelzang, 2013) has been applied in our simulation. A weighted analysis field is constructed by combining the scatterometer observations and a model prediction, and then the one lying closest to the analysis field is selected as the output solution. The problem is solved by minimizing a total cost function that combines both observation and NWP information: $J = J_{obs} + J_{NWP} = -2[\ln P(\overrightarrow{v_k}|\vec{x}) + \ln P(\vec{x}|\overrightarrow{x_b})]$,
where $\vec{x}$ is the true state of the surface wind field, $\overrightarrow{v_k}$ is the possible ambiguous wind solutions, $P(\overrightarrow{v_k}|\vec{x})$ is the conditional





probability of the $\overrightarrow{v_k}$ observed given $\vec{x}$, and $P(\vec{x}|\overrightarrow{x_b})$ is the conditional probability of surface wind field $\vec{x}$ given $\overrightarrow{x_b}$. Detail of the method can be found in [21,22,23].

A statistical comparison of 1st rank solution and 2DVAR performances of SCAT, WindRad, and SeaWinds are shown in Figure 10, Figure 11, Figure 12. For SCAT, the 1st rank solution wind field (Figure 10 (a)) shows poor retrieval quality in
5     the nadir and outer swath, while 2DVAR (Figure 10 (b)) effectively improves the retrieval results here; by the way a similar effect occurs for SeaWinds (Figure 12). The nadir swath of WindRad shows worse wind retrieval quality than the other parts of the swath (Figure 11 (a)) and 2DVAR is able to correct the false solutions appearing in the 1st rank solution.  Note that the rotation sampling pattern of WindRad is visible as regular disturbances along the swath. This implies that for the same WVC number, different sets of views are collected, depending on the phase of the antenna rotation, hence the wind retrieval
10     performance may vary, e.g., the expected MLE. One aspect needs to be noted: the 2DVAR with MSS works properly in our simulation, but the input wind field of the simulation is ECMWF model data, which is consistent with the 2DVAR background field. Even though a Gaussian-distributed geophysical noise has been added in the input wind field, it still might lead to a selection of wind solutions that tends to be close to the model wind field and hence somewhat overestimates performance.

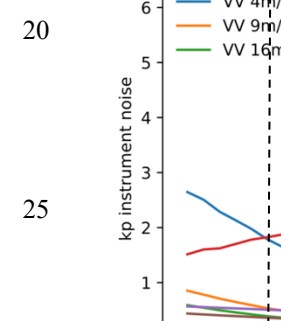
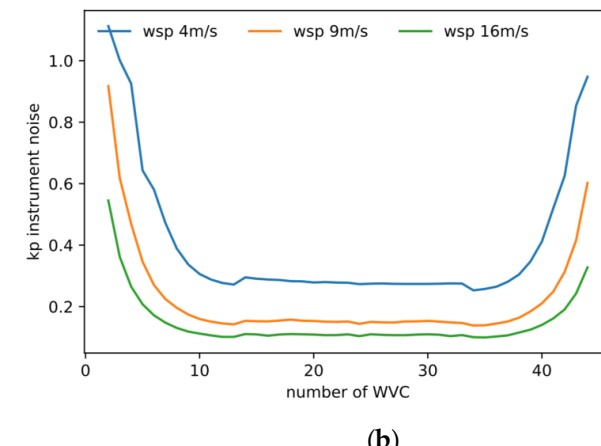

(a)                                     (b)

**Figure 7:SCAT instrument noise in ratio (1 is 100%) at 4m/s, 9m/s, and 16 m/s on (a) slice level; (b) WVC mean Kp.**





(a)

(b)

(c)

**Figure 8: WindRad instrument noise in ratio (1 is 100%) at 4m/s, 9m/s, and 16 m/s on (a) slice level of Ku-band; (b) slice level of**

30  **C-band (slices with SNR < 0.05 are excluded); (c) WVC mean Kp.**

35



**Figure 9:** SeaWinds instrument noise in ratio (1 is 100%) at 4m/s, 9m/s, and 16 m/s on (a) slice level of Ku-band HH pol; (b) slice level of Ku-band VV pol (slices with SNR < 0.05 are excluded); (c) WVC mean Kp.




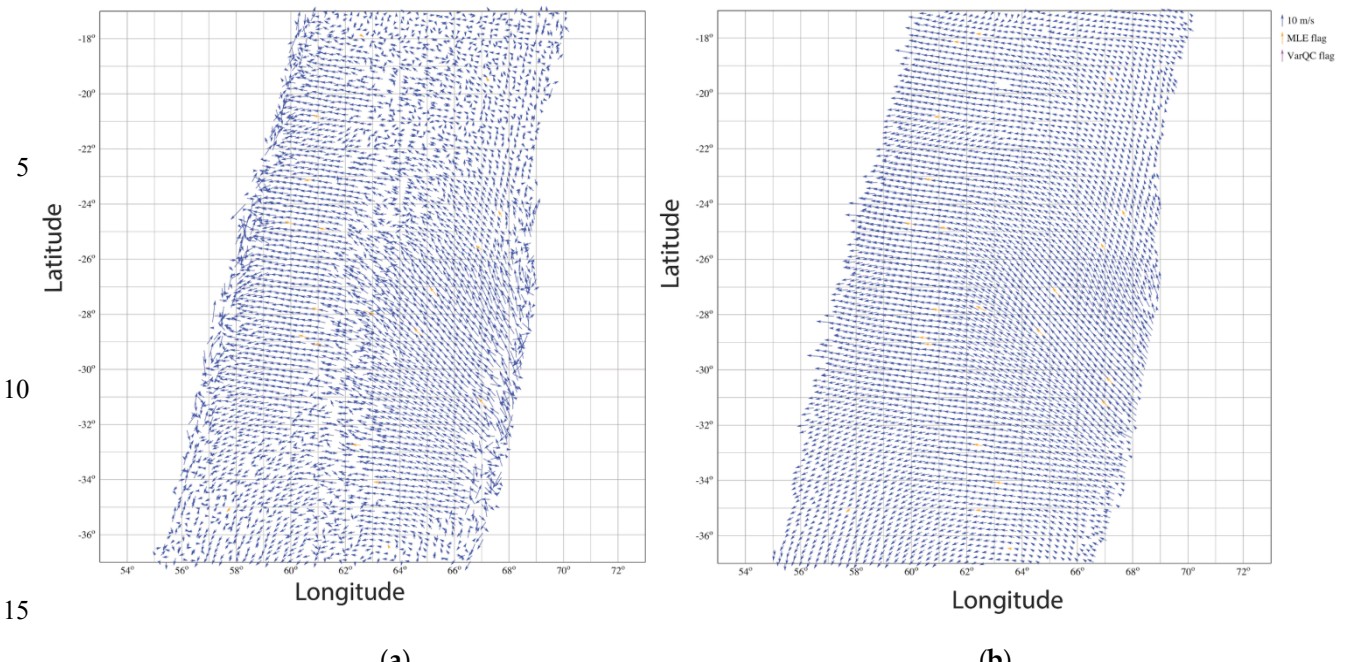

(a)          (b)

**Figure 10:** **SCAT retrieved wind field. (a) 1st rank solution; (b) 2DVAR result. The orange flags are artificial QC points and may be ignored.**

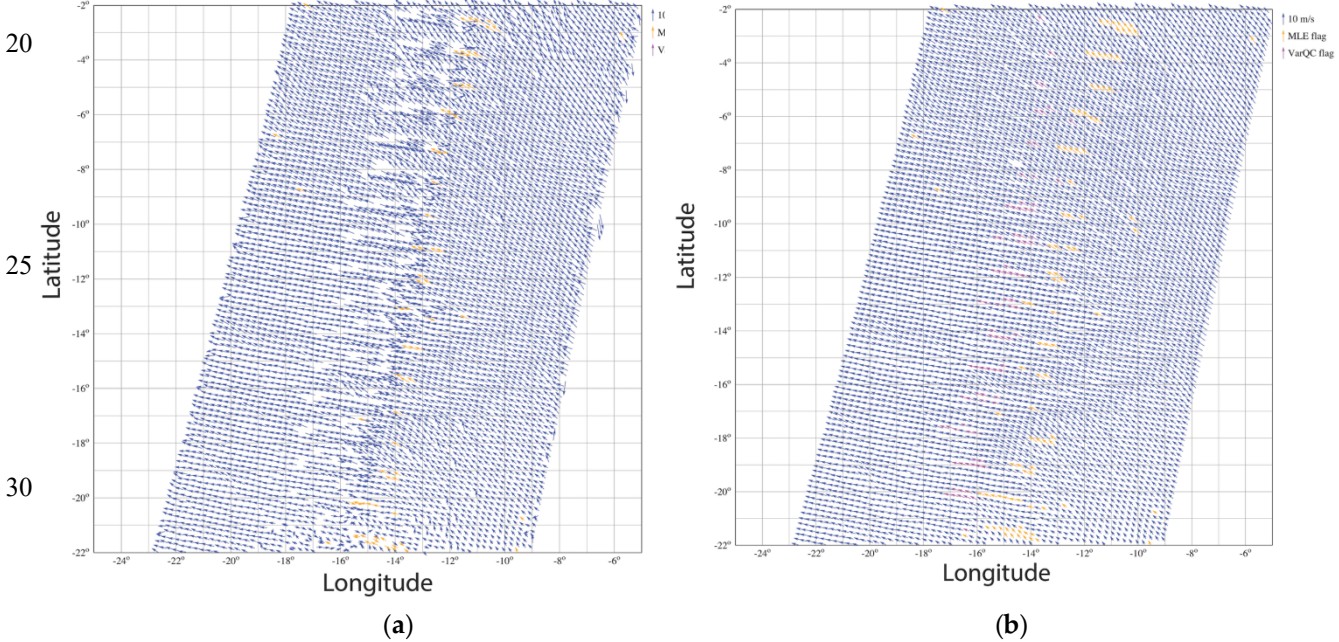

(a)          (b)

**Figure 11:** **WindRad retrieved wind field. (a) 1st rank solution; (b) 2DVAR result. The orange flags are artificial QC points and may be ignored.**





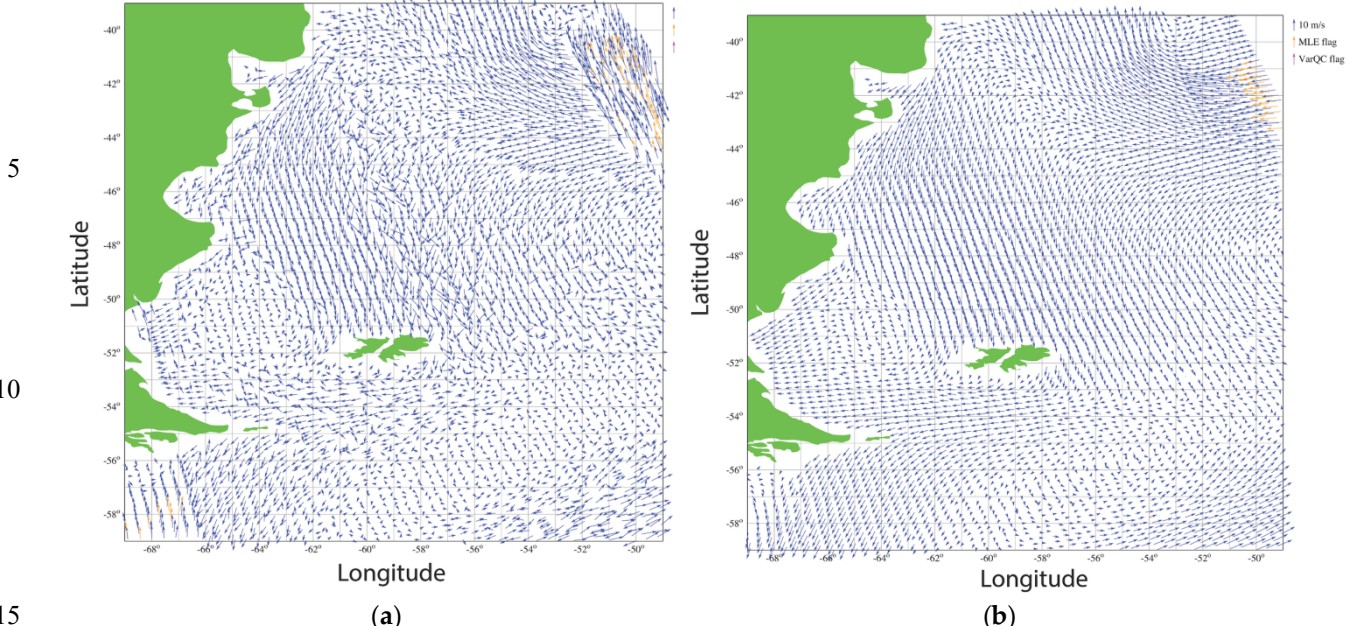

**Figure 12: SeaWinds retrieved wind field. (a) 1st rank solution; (b) 2DVAR result. The orange flags are artificial QC points and may be ignored.**

## 3. Simulation result and wind retrieval performance comparison

The simulation has been performed on SCAT, WindRad and SeaWinds with ECMWF model wind data (17th Dec, 2011) as input wind field. The swath widths for the three instruments are different, in order to make the following figures more comparable, the nadir WVCs of the three instruments are aligned.

### 3.1 Wind retrieval performance evaluation

### 3.1.1 Assessments with the input wind field

Four orbits of data on 2011-12-17 have been generated to be used for the wind retrieval simulation. The contoured histograms in Figure 13, Figure 14 and Figure 15 provide statistics of the wind speed, wind direction with respect to a wind blowing from the North, and wind components u (eastward) and v (northward) versus the variable "true" input wind field for SCAT, WindRad and SeaWinds. We note that opposing wind solutions will have opposite u and v signs and similar amplitude and therefore such common ambiguity appears as a cross pattern in the u and v histograms. This ambiguity is directly related to the main double harmonic dependency of the GMF (Wang et al., submitted, 2018).

For SCAT (Figure 13 (a)), the 1st rank solution of all WVCs across the swath are included. It shows rather poor statistics when compared with the input wind field. However, by simply excluding the WVCs located in the outer swath, the 1st rank solution quality improves substantially (Figure 13 (b)). The spread in the wind speeds is reduced and some derived false wind directions, which are shown as parallel and perpendicular lobes to the true value in the plots, are removed. When




the nadir-swath WVCs are also excluded (Figure 13 (c)), then the wind speed collocation statistics stay almost unchanged as compared to Figure 13b, while most of the false wind directions perpendicular to the true value are removed. This means that the outer swath contains the most ambiguous wind vector results, while the nadir swath ambiguities cause mainly wind direction errors.

Figure 14 (a) shows the 1st rank wind retrieval for WindRad with all WVCs and it shows much better statistics as compared to SCAT (Figure 13 (a)), due to twice the number of views in each WVC. Excluding outer WVCs (Figure 14 (b)) has less effect on the wind retrieval quality for WindRad than for SCAT. The retrieved wind speed shows a bit better statistic, but wind direction statistics stay almost unchanged, which means that the outer WVCs do not strongly increase the wind direction ambiguity. On the other hand, when we only exclude nadir WVCs (Figure 14c), the wind direction retrieval is

improved. The average wind speed bias is 0.42 m/s and the standard deviation of wind direction is 32.21° (Figure 14 (c)), while they are 0.51 m/s and 41.30° for Figure 14 (b), respectively. The last experiment shown for WindRad is to exclude both outer and nadir WVCs (Figure 14 (d)) with averaged wind speed bias of 0.44 m/s and standard deviation of wind direction of 35.61°. The largest performance improvement of WindRad occurs when excluding nadir WVCs. The outer swath mainly influences the wind speed retrieval skill, while the nadir swath provides wind direction ambiguity.

SeaWinds's outer swath contains only two views (fore-VV and aft-VV), and in order to process outer swath winds, each of these two views are split into two views based on their azimuth angle (four views in total in the end). Even though there are four views at the outer swath, the limited azimuth diversity leads to more ambiguous wind retrieval results (Figure 15). The wind retrieval quality of SeaWinds is the poorest one among these three instruments.

      The averaged wind retrieval statistics against the input wind field are dominated by the lack of ambiguity removal and

non-linearity. In practice these issues are successfully dealt with in the ambiguity removal step, using prior background information. In next section we determine Figures of Merit (FoM) to compare scatterometer performances with and without such prior information.





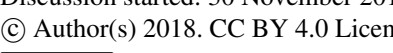

**Figure 13:** Contoured histograms of SCAT retrieved 1st rank wind solution versus input wind field for 4 orbits. (a) all WVCs within the swath; (b) excluding the WVCs in the outer swath, WVC number from 8 to 42 are included; (c) excluding the WVCs in the outer swath and nadir swath, WVC number from 8 to 17 and 26 to 42 are included. From (a) to (c), upper left: wind speed; upper right: wind direction; lower left: u component; lower right: v component. The contour lines are logarithmic.







**Figure 14:** Contoured histograms of WindRad retrieved 1st rank wind solution versus input wind field for 4 orbits. (a) all WVCs within the swath; (b) excluding the WVCs in the outer swath, WVC number from 20 to 60 are included; (c) excluding the WVCs in the nadir swath, WVC number from 35 to 45 are included; (d) excluding the WVCs in the outer swath and nadir swath, WVC number from 20 to 35 and 45 to 60 are included. The four figures in (a-d), upper left: wind speed; upper right: wind direction; lower left: u component; lower right: v component. The contour lines are logarithmic.



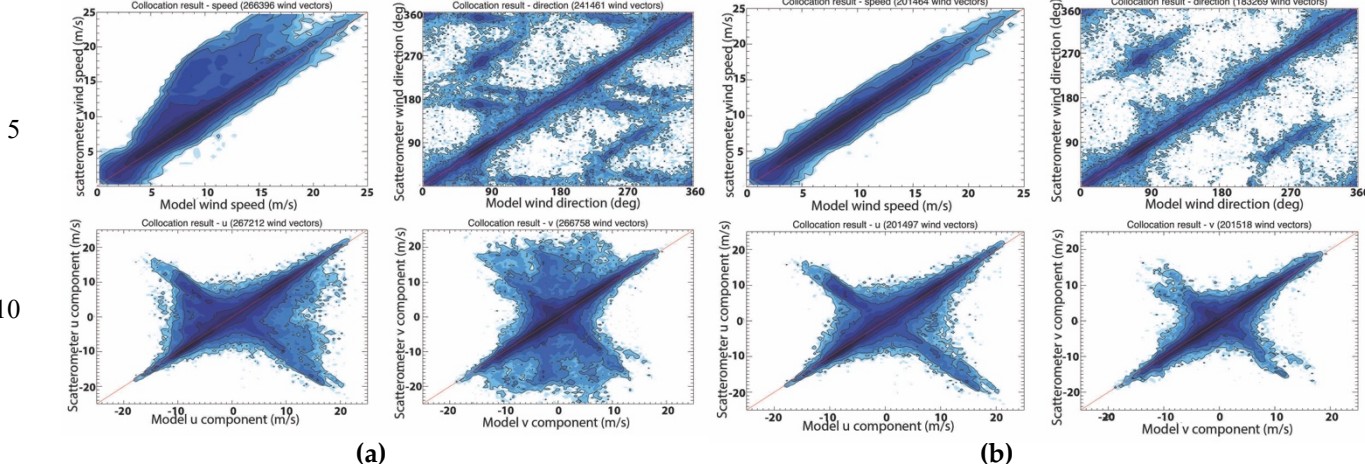

**(a)**                    **(b)**

**Figure 15: Contoured histograms of SeaWinds retrieved 1ˢᵗ rank wind solution versus input wind field for 4 orbits. (a) all WVCs within the swath; (b) excluding the WVCs in the outer swath, WVC numbers from 10 to 65 are included. From (a) to (b), upper left: wind speed; upper right: wind direction; lower left: u component; lower right: v component. The contour lines are logarithmic.**

**3.1.2 Figure of Merits**

The 1st rank solutions contain ambiguities and because the input wind field is the ECMWF model wind, but without spatially correlated error, it leads to a nearly perfect 2DVAR result, which is unrealistic. In order to further evaluate the wind retrieval performance, the ambiguity of the solutions may be statistically evaluated in the context of generally available background (NWP) information. Figures of Merit (FoM) are a set of parameters to evaluate the wind retrieval quality of different scatterometer concepts, taking into account imprecise, ambiguous and biased wind solutions. Three FoM, which are normalized wind Vector RMS error (VRMS), Ambiguity Susceptibility (AMBI) and systematic error (BIAS), are introduced here based on (Rivas et al., 2009). A brief description is given first.

The VRMS FoM is defined to quantify the ability of the scatterometer wind retrieval to handle ambiguous solutions with a priori NWP model information, such as in 2DVAR, but without actually simulating realistic spatially correlated errors. The input wind field to our simulation is considered as true winds (denoted with $\vec{v_t}$). VRMS quantifies the total simulated wind retrieval error with respect to $\vec{v_t}$. It is, however, calculated by down-weighting ambiguous wind vector solutions that are very distant from $\vec{v_t}$, since in practice it is easiest for 2DVAR and other applications to discard such solutions. The down-weighting involves the common prior knowledge in these applications, which is the general NWP background wind component uncertainty, denoted $\sigma_{NWP}$ and assumed equal for u and v. The ambiguous retrieved wind vector distribution, expressed in the wind probability $P_{obs}(\vec{v}|\vec{v_t})$, is multiplied by a Gaussian probability distribution $P_{NWP}(\vec{v} - \vec{v_t})$ centered at the input wind field and with a variance $\sigma^2_{NWP} \sim 5$ m2/s2 in both wind components. The VRMS FoM is subsequently obtained by normalizing this expression by the prior NWP VRMS error:



$$FoM_{VRMS} = \frac{RMS_{obs}}{RMS_{NWP}} \tag{2}$$

where $RMS_{obs} = \left(\sqrt{\int |\vec{v} - \vec{v_t}|^2 P_{obs}(\vec{v}|\vec{v_t}) \times P_{NWP}(\vec{v} - \vec{v_t}) d^2v}\right)$ and $RMS_{NWP} = \left(\sqrt{\int |\vec{v} - \vec{v_t}|^2 P_{NWP}(\vec{v} - \vec{v_t}) d^2v}\right) =$

$\sqrt{2}\sigma_{NWP}$. VRMS quantifies the wind solution's relative RMS about the true wind with respect to the general prior background uncertainty. If its value is 1, then the wind retrieval failed to provide new and useful information in the wind field, i.e., corresponding to $P_{obs}(\vec{v}|\vec{v_t})$ =constant.

5      On the other hand, AMBI is defined to quantify the ability of the scatterometer and its processing to handle ambiguous solutions without a priori NWP model information. It is a ratio of the wind solution output falling outside the general prior wind field constraint, relative to the output falling inside the prior wind field constraint. The lower the ratio, the better (3), where $P_{NWP,max}$ is the maximum probability of $P_{NWP}(\vec{v} - \vec{v_t})$.

$$FoM_{AMBI} = \frac{\int P_{obs}(\vec{v}|\vec{v_t}) \times \left(P_{NWP,max} - P_{NWP}(\vec{v} - \vec{v_t})\right) d^2v}{\int P_{obs}(\vec{v}|\vec{v_t}) \times P_{NWP}(\vec{v} - \vec{v_t}) d^2v} \tag{3}$$

BIAS quantifies the systematic vector wind bias, again in the context of the background prior, which is the shift of the average location of the output wind solution away from the location of the prior wind caused by skewness in the output wind solutions (4).

$$FoM_{BIAS} = \int (\vec{v} - \vec{v_t}) \cdot P_{obs}(\vec{v}|\vec{v_t}) \times P_{NWP}(\vec{v} - \vec{v_t}) d^2v \tag{4}$$

The wind retrieval is a non-linear problem and the output wind error depends on the true wind vector (wind speed and direction distribution). In order to minimize this dependence, the calculated FoMs are averaged over a climatology of wind inputs with uniformly distributed directions and wind speeds (3 m/s to 16 m/s) following a Weibull distribution with a maximum around 8 m/s (Liu et al., 2008). The input wind speeds are from 3 to 16 m/s with steps of 1 m/s, and the input wind directions are from 0 to 360 degrees with steps of 10 degrees. Each wind speed and wind direction combination contains the equivalent number of WVCs from the same orbit.

Figure 16 gives the three FoM comparisons of SCAT, WindRad and SeaWinds as a function of the WVC positions in the swath. Overall, the wind retrieval performance of the rotating fan-beam instruments is better than the pencil-beam instrument, while the outer and nadir swaths of the three instrument types yield a poorer performance than the sweet swaths. The outer swath of SeaWinds only has two independent views, which result in very ambiguous winds and the worst simulated wind retrieval quality. The wind retrieval quality across the swath strongly depends on the location of the WVC; it degrades substantially in the outer and nadir swaths as expected. The outer swath has worse quality than the nadir swath for both SCAT and SeaWinds, whereas these two regions are showing comparable wind retrieval quality for WindRad. Although the number of views in the sweet swath for WindRad is twice the number for SCAT (Figure 6), the wind retrieval





quality is not improved as expected, but shows very similar quality to SCAT. The elevated values for AMBI and BIAS indicate that, despite the high number of views, the wind retrieval tends to be not well determined and slightly non-linear. At the same time, the quality in the outer swath of WindRad shows very pronounced improvement with respect to SCAT, due to the increased number of available views.

Figure 17 illustrates the VRMS as a function of wind direction and WVC location at 9 m/s wind speed. The wind retrieval performance across the swath for all wind directions gives the same pattern as described above with some modulations at different wind direction. There is one different feature occurring for WindRad. The VRMS at nadir swath shows higher values than in the outer swath, which is opposite to SCAT and SeaWinds. AMBI and BIAS (not shown here) have similar patterns as VRMS.

**(a)**

**(b)**

**(c)**

**Figure 16:  FoM results of SCAT, WindRad, and SeaWinds. (a) VRMS comparison; (b) AMBI comparison;(c) BIAS comparison.**



**Figure 17: FoM VRMS map as a function of across-track location and wind direction (wind speed is 9 m/s). (a) SCAT; (b) SeaWinds; (c) WindRad.**

### 3.1.3 Wind direction bias

Wind direction bias between the wind retrieval result (2DVAR result) and the ECMWF model wind has been evaluated as a function of WVC and relative wind direction (15 orbits are included). The relative wind direction means the retrieved wind direction relative to the satellite motion direction. In this evaluation, we are able to see the wind direction bias with respect to the true direction at all the WVCs (Figure 18). No matter the biases are negative or positive, both signs indicate that the wind directions have a tendency to be closer to the satellite motion direction and if the wind direction bias is averaged over all the relative wind directions, a small value of the bias remains.





SeaWinds gives stronger bias both on the outer swath and nadir swath (Figure 18 (b)), while the nadir swath of SCAT gives weaker bias comparing to the outer swath due to the increased number of views (Figure 18 (a)). For WindRad, when the retrieved wind direction is close to satellite motion direction (relative wind direction is 0), it shows rather strong negative and positive bias, but the non-biased area for WindRad is larger than it is in SCAT and SeaWinds. This phenomena can also

5    be observed with real data from SCATSAT (Wang et al., submitted, 2018). This retrieved wind direction preference might be caused by the retrieval method.

**(a)**

**(b)**

**(c)**

**Figure 18: Wind direction bias between wind retrieval result and true wind as a function of WVC number and the relative wind direction (y-axis, the retrieved wind direction relative to the satellite motion direction; color scale is consistent for easy comparison, all the values outside [-20, 20] are marked as dark blue and dark red). (a) SCAT; (b) SeaWinds; (c) WindRad.**



## 4. Discussion

Our results confirm that the wind retrieval quality of Ku-band rotating-beam scatterometers (rotating fan-beam and rotating pencil-beam) varies according to the location of the WVCs across the swath, and the outer and nadir swaths show generally lower wind retrieval skill than the sweet swath. The wind retrieval comparisons suggest that WindRad gives the best wind retrieval quality of the three scatterometer types, although its increased number of views not always lead to further wind retrieval quality improvement, particularly in the sweet swath.

Prior work on SCAT and WindRad mainly focused on the instrument configuration choices and tests with various settings of pulse frequency, polarization, rotating speed, transmitted peak power, etc. (Lin et al., 2000a, 2002; Lin and Dong, 2011). The main wind retrieval performance characteristics of these two-rotating fan-beam scatterometers are in line with our study with respect to sweet swath, nadir and outer swath performance. SCAT has been compared with SeaWinds (Lin and Dong, 2011) in an end-to-end simulation and now we add a cross comparison among SCAT, WindRad and SeaWinds in the same simulation framework as before.

WindRad shows distinguished and new wind retrieval features in our study with respect to SCAT and SeaWinds. The outer swaths of SCAT and SeaWinds clearly provide the worst wind retrieval skill as compared to nadir, but for WindRad this does not occur. WindRad gives very similar wind retrieval quality in the nadir and outer swaths according to its FoM (Figure 16). However, the spread of the FoM values are largest in the inner swath (Figure 17) and excluding the inner swath leads to a better wind retrieval instead of excluding the outer swath. This is opposite to SCAT and SeaWinds, where performance increases most when excluding the outer swath.

Increasing the number of views and azimuth diversity leads to an improved wind retrieval. While this is generally true, comparing SCAT and WindRad, the increased number of WindRad views on their own provide a strong improvement in the outer swath, but it appears not effective in the sweet and nadir swath. The azimuths in the nadir swath are always either looking forward of the satellite track or looking backward. The increased views in this case will still be with similar azimuths, and these are not effective to improve the azimuth diversity. In the sweet swath, there are up to 18 views in the WVCs leading to a more diverse observation geometry. However, the incidence angle range of SCAT, with almost 25 degrees, is much broader than that from WindRad at about 10 degrees, where, moreover, the minimum SCAT incidence angle at 25 degrees is about 10 degrees lower than that from WindRad. This implies that, while moving away from nadir, the azimuth diversity of SCAT increases much faster than that of WindRad, hence the steep performance increase for SCAT when moving away from nadir. On the other side, the many channels on WindRad and its fan beam add a lot of additional views and azimuth diversity near the outer swath, as compared to SCAT and SeaWinds, hence the outstanding outer swath performance of WindRad. We found that the number of slices located in the outer swath is the least and the geometrically unbalanced $\sigma°$ distribution within the views of a WVC is one of reasons for the low retrieval quality. The instrument noise also contributes, and it is lower for WindRad than SCAT on average. All in all, the wind retrieval is substantially better in the outer swath for WindRad. We also note that SCAT is essentially providing reference wind information for the CFOSAT small-incidence wave instrument SWIM and as such well designed for this task.



## 5. Conclusions

In summary, we have presented and assessed a generic simulation framework, which has been adapted to all existing rotating-beam scatterometer types. The representative set of SCAT, WindRad and SeaWinds is chosen to evaluate the wind retrieval performance of the rotating scatterometers using Ku-band. The wind retrieval quality strongly depends on the location of the WVCs across the swath, and the sweet swath shows the most favorable geometries for wind retrieval. Among the more unfavorable outer and nadir swath regions, SCAT and SeaWinds both perform best in the nadir swath, while WindRad rather substantially improves the outer swath wind retrieval. On the other side, WindRad's nadir swath region with lower wind retrieval quality is larger than its outer swath region, while SCAT and SeaWinds have a relatively large outer swath region with degraded quality. The outer swath of SCAT implies both wind speed and wind direction retrieval problems, while for WindRad only wind speed retrieval is affected. Although rotating fan-beam scatterometers, particularly SCAT, much improve nadir performance with respect to SeaWinds. the nadir swath shows still significant wind direction ambiguity for both SCAT and WindRad.

The increased number of views in the nadir and sweet swath for WindRad does not lead to an improved wind retrieval, but it shows a saturation effect and stays relatively similar to SCAT. The retrieved wind direction has a tendency towards the satellite motion direction for all three instruments, which is related to the retrieval procedure. Rain effect is not taken into consideration, so the rain disturbance in Ku-band and the advantage of the C-band on WindRad cannot be shown here. Further studies may focus on this aspect.

To facilitate good quality collocations with the CFOSAT SWIM instrument, the SCAT design is clearly focused on an optimal performance close to nadir and employs small incidence angles, combined with a large incidence angle range. This facilitates the availability of additional views near nadir with enhanced azimuth and incidence angle diversity. On the other hand, WindRad's most useful complement is clearly its dual frequency capability, providing many views in the outer swath, where excellent performance is obtained according to our simulations.

This simulation allows us to further investigate the true resolution of the instruments before their launch and also the non-overlap of the views in a WVC, which contributes to the geophysical noise. The WVC size is not the true spatial resolution and neither the true representation of the contributing views, which depend on the spatial response function of each sample, how these are aggregated into a view and which views contribute to the WVC (Vogelzang et al., 2017; Vogelzang and Stoffelen, 2017). For rotating-beam scatterometers the sampling and hence wind retrieval characteristics vary potentially both by across-track and along-track WVC, which may be further investigated. Such development may much help users interested in coastal winds.

Our simulation does not consider the rain effect. Ku-band ocean returns are affected by rain, and moderate and heavy rain will certainly degrade the wind retrieval. At KNMI, we use the wind retrieval MLE for rain screening of Ku-band systems, much aided by MSS 2DVAR. This successful methodology developed for SeaWinds will also be attempted for CFOSAT. On the other hand, C-band backscatter is much less sensitive to rain and included in WindRad. This advantage of WindRad should be further investigated, e.g., by using collocated measurements of Ku-band and C-band scatterometers.





The broad user community looks forward to an increased temporal sampling of the ocean surface with scatterometer winds, such as with both WindRad and SCAT, that will be useful contributions to the global ocean surface vector winds virtual constellation.

**Acknowledgments:** This work was supported by the project of 'the development and provision of scatterometer wind processing software and wind products for the China-France Oceanography SATellite (CFOSAT)' between CNES (Centre National d'etudes Spatiales) and KNMI (Royal Netherlands Meteorological Institute), supported by the EUMETSAT Ocean and Sea Ice Satellite Application Facility (OSI SAF).

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
