# Peer review of "A Generalized Simulation Capability for Rotating Beam Scatterometers"

_Atmospheric Measurement Techniques, 2018_

## Editor Comment (EC1) · Andreas Richter (Editor) · 12 Feb 2019

Referee #1 submitted a detailed review of the manuscript already in the quick-access review phase. Many of the points have already been addressed by the authors when submitting the version of the manuscript now available for discussion. The remaining point is repeated below. As there will be no second review from referee #1, this is the only point which needs to be addressed with respect to that review.

========================================================================

Open review of Li et al., "Simulation of rotating Ku-band wind scatterometers and wind retrieval performance comparison"

This paper compares scatterometer figure of merits (FoMs) for several different Ku-

band scatterometer geometries, specifically the rotating pencil-beam SeaWinds on QuikSCAT, and the rotating fan-beam WindRad on FY-3E and SCAT on CFOSAT. The paper purports to develop a unified framework for analyzing and comparing the performance of these systems where, based pm the FoMs used, WindRad has the best performance, followed by SCAT and SeaWinds. The authors find that the FoMs are sensitive to the number of the "views" (also referred to as "flavors" in the literature) of the normalised radar backscatter sigma-0. Each "view" corresponds to a different azimuth angle and/or incidence angle and/or polarisation so the set of measurement views span a diversity of azimuth angle, incidence angles, and polarisations. They find that the FoMs improve with increasing number of "views" until a saturation is reached.

Overall, the paper is well written well and clear. A few corrections and additions could substantially improve the paper.

One thing not addressed in the paper is the relationship between the number of views and their diversity. It is not the number of views that matter but their diversity. It should be noted that views that have similar geometry and polarization do not really add to the measurement diversity, and so do not add information to the wind retrieval process. Such measurements can be grouped (averaged) into a "super" view that is treated as a single view without affecting the wind retrieval. In fact, this is frequently done when processing rotating pencil-beam scatterometers: sigma-0 measurements with similar views ("flavors" in the QuikSCAT literature) are often averaged to simply processing. For this paper there must be a way to quantify the diversity of the views (other than just using their number). This should be used to investigate how the diversity contributes the FOMs' value. Bottom line: adding additional views that do not have distinctly different geometry (i.e., that do not really add to the diversity) cannot be expected to improve the wind retrieval. Such additional measurements are essentially only improving the effective SNR of the class (super view) of similar views rather than adding new geometric information need to improve the wind retrieval.

---

## Referee Comment (RC1) · Anonymous Referee #2 · 14 Feb 2019

This is an interesting paper which 1) describes a simulation and FoM methodology, and 2) uses this methodology to evaluate the relative performance of three recent specific scatterometer designs. This paper would be even more valuable to the community if it described the system parameters of SCAT and WindRad in a little more detail (as suggested below). I also have some specific questions, comments, and suggestions in the text.

Page 3, Line 3: Question: Have either SCAT or WindRad been launched? Are there any references to their design and on-orbit performance?

Page 5, Tables 1,2,and 3: {

Correction: What is currently listed as "antenna bandwidth" in the table is perhaps more

appropriately termed "center frequency."

Recommendation: In Tables 1-3 include the actual TRANSMIT BANDWIDTH in the table. This would be extremely valuable for the readers to understand how many independent range looks are available for each slice measurement. (For instance, from the literature SeaWinds has a transmit bandwidth of 375 kHz.).

Recommendation: Specifically state the number of independent looks (not views) for each slice.

Recommendation: In Tables 1-3 add what the Noise Equivalent sigma-0 is for each system. Perhaps it is actually a range of values depending on the specific slice position within the antenna footprint on the ground. }

Recommendation: Add a new diagram/figure showing how each antenna footprint is "sliced" using range processing. What are the dimensions of the individual slices on the ground? What is the overall spatial resolution of each system?

Page 7, Lines 1-5: Comment: The authors are correct in indicating that the coefficients A, B, and C are a function of the precise detection scheme. The approximations for A, B, and C given in the paper are identical to those derived for SeaWinds, which uses a deramp detection of the chirped bandwidth and then frequency filtering to obtain each slice. It is unclear whether they are applicable to the SCAT or WindRad cases because the dection scheme is not specified.

Question: I don't understand what the statement "The distribution of Bs on each slice in one pulse is assigned according to the antenna gain pattern of the pulse" means.

Page 7, Figure 3: Question: Is there any error term in the simulation for the radiometric calibration accuracy? Radiometric calibration accuracy is another factor important in scatterometry. What is the assumed or achieved radiometric calibration accuracy for SCAT, WindRad and SeaWinds?

Page 9, Figure 6: Question: What are you defining as being a "view." Specifically for

[Figure]

SeaWinds, my understanding is that for the outer WVC's, there are measurements that occur from multiple azimuth angles for multiple antenna rotation, although it is a very small range of azimuth angle variation). For instance, in the paper "Point-Wise Wind Retrieval and Ambiguity Removal Improvements for the QuikSCAT Climatological Data Set," A.G. Fore et. al., IEEE Trans. on Geosci. and Remote Sensing, VOL 52, No. 1, January 2014, it shows a distinct "saddle shaped" distribution of "composites" as a function of WVC, not a flat distribution as shown in the author's Figure 6. What is the difference between "composites" in the above paper and "views" in this paper?

Page 10, Lines 4,5: Comment: The line that reads ". . . the Kpc on the WVC level is derived by averaging the Kpc for all the views in the corresponding WVC." Wouldn't the Kpc instead be actually reduced when all the views of included together? As multiple s0's from different views are averaged, wouldn't the aggregate Kpc go down?

Pages 14 and 15: Question: Figures 10 and 11 appear to be a model simulation output whereas Figure 12 is an actual SeaWinds measured wind field (?).

Page 20, Line 20: Comment: The statement "Overall the wind retrieval performance of the rotating fan-beam instruments is better than the pencil-beam instrument." Clearly more "views" are better than fewer views, but the number of looks is also important. This conclusion may be the case for this specific pencil-beam scatterometer (Sea-Winds) with its relatively small bandwidth and low number of looks per slice, but a pencil beam scatterometer with a higher gain and/or higher transmit bandwidth could potentially compensate for the lack of views. There may be a trade-off here.

Page 25, Conclusions: Comment: One aspect that I find seriously missing in this paper is the acknowledgement that SeaWinds (as well as SCAT and WindRad maybe?) have already been operating in orbit. In the case of SeaWinds, there is an approximately 10 year data record that has been extensively evaluated. Yet the actual performance of the scatterometers on actual wind fields is not compared to the model simulation results. It seems that this would be a good means of establishing the validity of the

model, particularly with regards to evaluating the "geophysical noise." The chances are good, I would guess, that the model performance is actually better than that observed in the real world in all cases. Thus the model/simulation evaluation might best be said to be an evaluation of "relative performance potential" amongst various scatterometer designs as opposed to actual real world performance.

---

## Author Comment (AC2) · 16 Mar 2019

Thank you very much for your comment. Indeed, the increase of the views without increasing the diversity is not effective in improving the wind retrieval. It is possible to do another analysis of the geometry diversity of the views in different WVC and check their influence to the wind retrieval result.

---

## Author Response (AR1)

**Referee #1:**

*One thing not addressed in the paper is the relationship between the number of views and their diversity. It is not the number*
5  *of views that matter but their diversity. It should be noted that views that have similar geometry and polarization do not*
*really add to the measurement diversity, and so do not add information to the wind retrieval process. Such measurements*
*can be grouped (averaged) into a "super" view that is treated as a single view without affecting the wind retrieval. In fact,*
*this is frequently done when processing rotating pencil-beam scatterometers: sigma-0 measurements with similar views*
*("flavors" in the QuikSCAT literature) are often averaged to simply processing. For this paper there must be a way to*
10  *quantify the diversity of the views (other than just using their number). This should be used to investigate how the diversity*
*contributes the FOMs' value. Bottom line: adding additional views that do not have distinctly different geometry (i.e., that*
*do not really add to the diversity) cannot be expected to improve the wind retrieval. Such additional measurements are*
*essentially only improving the effective SNR of the class (super view) of similar views rather than adding new geometric*
*information need to improve the wind retrieval.*

An experiment has been done to verify the influence of the views with similar geometry. The following text has been added
in discussion session:

One question is raised here: is the azimuth diversity the major influencer in the wind retrieval quality or the number of
20  views with similar azimuth also give added value in the wind retrieval process? The views with similar azimuth geometry
are averaged together into one super view. This influences most of the nadir swath since the views are with mainly forward
and backward azimuths. Figure 23 shows that the views with similar azimuth angle in one of the nadir WVCs are aggregated
into one view and the views for the nadir swath becomes mainly one forward and one backward. The wind retrieval results
(Figure 24) in comparison with the normal wind retrieval results (Figure 17) show that the wind retrieval result with all the
25  WVCs looks very similar, which seems that the views with similar azimuth angle do not have much added value onto the
wind retrieval process, but when the outer swath WVCs are excluded from the result, wind speed and wind direction (Figure
24 (b)) are both worse than Figure 17 (b) as well as when excluding the nadir swath. In Figure 17 (b) and (c), the nadir swath
does not influence the wind speed retrieval much, while here (Figure 24 (b) and (c)) the wind speed and wind direction
retrieval results are obviously improved after excluding nadir swath. This means the reduction of the number of views in the
30  nadir swath after aggregating the views with similar azimuth angle into one view leads to a worse wind retrieval quality.

**Referee #2:**

*Page 3, Line 3: Question: Have either SCAT or WindRad been launched? Are there any references to their design and on-orbit performance?*

SCAT has been launched on 29[th] Oct 2018, but WindRad has not been launched yet and the plan is this year. The reference of SCAT design and on-orbit performance is at session 2.1 'The RFSCAT characteristics have been studied and assessed by Lin et al. (Lin et al., 2000a, 2002)'. The reference for WindRad is added in the manuscript:

The scatterometer (referred to as SCAT from now on) onboard CFOSAT (China-France Oceanography SATellite) and WindRad (Chinese Wind Radar on FY-3E) belong to this type of scatterometer and are planned to be launched in 2018 and 2019 (Dou et al., 2014).

*Page 5, Tables 1, 2, and 3*

*Correction: What is currently listed as "antenna bandwidth" in the table is perhaps more appropriately termed "center frequency."*

Yes, it has been corrected (see table 1, 2, and 3).

*Recommendation: In Tables 1-3 include the actual TRANSMIT BANDWIDTH in the table. This would be extremely valuable for the readers to understand how many independent range looks are available for each slice measurement. (For instance, from the literature SeaWinds has a transmit bandwidth of 375 kHz.).*

They are added in Tables 1-3.

*Recommendation: Specifically state the number of independent looks (not views) for each slice.*

The technical parameters are not able to get for CFOSAT and WindRad (they are not allowed to release to public from the China part). The simulation does not use the number of independent looks for each slice, we simplified the simulation by cutting slices the same size from the pulse, so unfortunately, we do not have the number of independent looks for each slice.

*Recommendation: In Tables 1-3 add what the Noise Equivalent sigma-0 is for each system. Perhaps it is actually a range of values depending on the specific slice position within the antenna footprint on the ground.*

It is indeed a range of values depending on the specific slice position within the antenna footprint. The NESZ plots for all the antennas of each scatterometer are added in the manuscript after the Tables 1-3.

Text are added in 2.1:

NESZ (Noise Equivalent Sigma-Zero) is a range of values depending on the specific slice position within the antenna footprint on the ground. Figure 3, Figure 4, and Figure 5 give the NESZ distribution as a function of the slice number for SCAT, WindRad and SeaWinds.

*Recommendation: Add a new diagram/figure showing how each antenna footprint is "sliced" using range processing. What are the dimensions of the individual slices on the ground? What is the overall spatial resolution of each system?*

In the simulation, the slicing is simplified by cutting the pulse into the equal length slices, so maybe it is not necessary to have a diagram, and text is added in session 2.3 'In order to simplify the simulation procedure, the pulse is cut into equal size slices.'

It is two dimensions of the individual slices on the ground. The overall spatial resolutions are CFOSAT 10km, WindRad (C-band 20km, Ku-band 10km), SeaWinds 25km.

*Page 7, Lines 1-5: Comment: The authors are correct in indicating that the coefficients A, B, and C are a function of the precise detection scheme. The approximations for A, B, and C given in the paper are identical to those derived for SeaWinds, which uses a deramp detection of the chirped bandwidth and then frequency filtering to obtain each slice. It is unclear whether they are applicable to the SCAT or WindRad cases because the detection scheme is not specified.*

*Question: I don't understand what the statement "The distribution of Bs on each slice in one pulse is assigned according to the antenna gain pattern of the pulse" means.*

We do not have the access to the detection scheme of CFOSAT and WindRad, so the approximation is applied here. The approximation is to give distribute the pulse bandwidth onto slices with the distribution pattern of antenna gain pattern. The antenna gain pattern has the feature that peak at the center and gradually decreasing to the sides as a function of the distance to the center. So, the slice_bandwidth = pulse_bandwith * antenna_pattern.

*Page 9, Figure 6: Question: What are you defining as being a "view." Specifically, for SeaWinds, my understanding is that for the outer WVC's, there are measurements that occur from multiple azimuth angles for multiple antenna rotation, although it is a very small range of azimuth angle variation). For instance, in the paper "Point-Wise Wind Retrieval and*
5   *Ambiguity Removal Improvements for the QuikSCAT Climatological Data Set," A.G. Fore et. al., IEEE Trans. on Geosci. and Remote Sensing, VOL 52, No. 1, January 2014, it shows a distinct "saddle shaped" distribution of "composites" as a function of WVC, not a flat distribution as shown in the author's Figure 6. What is the difference between "composites" in the above paper and "views" in this paper?*

10   For SeaWinds, the classic way of defining the views is that the slices in one WVC are classified by fore/after beam first and then within the inner swath, the slices are classified by their polarization into two views (VV and HH), so there are four views at the inner swath. The outer swath only has one polarization, so it only has two views (fore/after beams). The definition of the composites in the paper (2014) is not very clear, they used a new method to aggregate the slices into one WVC, but the next step of grouping into views is not clearly described, so I would think the definition of composites is
15   different from the views in my paper.

*Page 10, Lines 4,5: Comment: The line that reads ". . . the Kpc on the WVC level is derived by averaging the Kpc for all the views in the corresponding WVC." Wouldn't the Kpc instead be actually reduced when all the views of included together? As multiple s0's from different views are averaged, wouldn't the aggregate Kpc go down?*

Yes, the Kpc will go down with the averaging all views or slices together and to reduce Kpc is important as well.

*Pages 14 and 15: Question: Figures 10 and 11 appear to be a model simulation output whereas Figure 12 is an actual*
25   *SeaWinds measured wind field (?).*

They are all simulated output. The purpose of these figures is to show that 2DVAR has been working in the simulation, so where to select the area is not important.

30   *Page 20, Line 20: Comment: The statement "Overall the wind retrieval performance of the rotating fan-beam instruments is better than the pencil-beam instrument." Clearly more "views" are better than fewer views, but the number of looks is also important. This conclusion may be the case for this specific pencil-beam scatterometer (SeaWinds) with its relatively small bandwidth and low number of looks per slice, but a pencil beam scatterometer with a higher gain and/or higher transmit bandwidth could potentially compensate for the lack of views. There may be a trade-off here.*

Yes, this trade-off is possible. In general, the number of views and the geometry diversity of the views play a major part in the wind retrieval quality, and the higher transmit bandwidth will lead to a high number of looks per slice thus leads to a better retrieval result. The number of views of rotating fan beam is significantly larger than pencil beam, so this trade-off might play a less important role here.

*Page 25, Conclusions: Comment: One aspect that I find seriously missing in this paper is the acknowledgement that SeaWinds (as well as SCAT and WindRad maybe?) have already been operating in orbit. In the case of SeaWinds, there is an approximately 10-year data record that has been extensively evaluated. Yet the actual performance of the scatterometers on actual wind fields is not compared to the model simulation results. It seems that this would be a good means of establishing the validity of the model, particularly with regards to evaluating the "geophysical noise." The chances are good, I would guess, that the model performance is actually better than that observed in the real world in all cases. Thus the model/simulation evaluation might best be said to be an evaluation of "relative performance potential" amongst various scatterometer designs as opposed to actual real world performance.*

The SCAT is in the orbit now (still in calibration process, not ready for release), and WindRad is not launched yet. The validation between simulated SeaWinds and actual SeaWinds data is a good evaluation for the simulation system and it has been done now, the following texts are added:

[revised manuscript text omitted]
\int|\boldsymbol{v} - \boldsymbol{v}_{\tilde{t}}|^2 P_{obs}(\boldsymbol{v}|\boldsymbol{v}_{\tilde{t}}) \times P_{NWP}(\boldsymbol{v} - \boldsymbol{v}_{\tilde{t}})d^2\boldsymbol{v}}\right)$ and $RMS_{NWP} = \left(\sqrt{\int\int|\boldsymbol{v} - \boldsymbol{v}_{\tilde{t}}|^2 P_{NWP}(\boldsymbol{v} - \boldsymbol{v}_{\tilde{t}})d^2\boldsymbol{v}}\right) =$ $\sqrt{2}\sigma_{NWP}$. VRMS quantifies the wind solution's relative RMS about the true wind with respect to the general prior background uncertainty. If its value is 1, then the wind retrieval failed to provide new and useful information in the wind field, i.e., corresponding to $P_{obs}(\boldsymbol{v}|\boldsymbol{v}_{\tilde{t}})$ =constant.

5    On the other hand, AMBI is defined to quantify the ability of the scatterometer and its processing to handle ambiguous solutions without a priori NWP model information. It is a ratio of the wind solution output falling outside the general prior wind field constraint, relative to the output falling inside the prior wind field constraint. The lower the ratio, the better (3), where $P_{NWP,max}$ is the maximum probability of $P_{NWP}(\boldsymbol{v} - \boldsymbol{v}_{\tilde{t}})$.

$$FoM_{AMBI} = \frac{\int P_{obs}(\boldsymbol{v}|\boldsymbol{v}_{\tilde{t}}) \times \left(P_{NWP,max} - P_{NWP}(\boldsymbol{v} - \boldsymbol{v}_{\tilde{t}})\right)d^2\boldsymbol{v}}{\int P_{obs}(\boldsymbol{v}|\boldsymbol{v}_{\tilde{t}}) \times P_{NWP}(\boldsymbol{v} - \boldsymbol{v}_{\tilde{t}})d^2\boldsymbol{v}} \tag{3}$$

BIAS quantifies the systematic vector wind bias, again in the context of the background prior, which is the shift of the
10   average location of the output wind solution away from the location of the prior wind caused by skewness in the output wind solutions (4).

$$FoM_{BIAS} = \int (\boldsymbol{v} - \boldsymbol{v}_{\tilde{t}}) \cdot P_{obs}(\boldsymbol{v}|\boldsymbol{v}_{\tilde{t}}) \times P_{NWP}(\boldsymbol{v} - \boldsymbol{v}_{\tilde{t}})d^2\boldsymbol{v} \
[revised manuscript text omitted]

---

## Author Response (AR2)

Response to the reviews

1) The new section on the influence of the views with similar geometry lacks clarity and needs some editing. Please rephrase to make it more clear what exactly has been tested, why it has been tested, what was found and what the conclusions are.

It has been updated.

One question is raised here: the relationship between the number of views and their diversity. It seems that the views with similar azimuth angle and the same polarization do not really add diversity into the system, which means they will not improve the wind retrieval result. Such views in the rotating pencil-beam scatterometer are averaged into one view without affecting the wind retrieval. Will we get the same effect for rotating fan-beam scatterometers? In order to investigate this, an experiment was done for SCAT where the views with azimuth angle difference within five degrees are averaged together into one super view, and the other parts of the wind retrieval are kept the same. This average procedure influences the nadir swath strongest since there are only forward and backward azimuth angles (Figure 23) and the number of views has been reduced according to the similarity of the azimuth angle. The same wind retrieval procedure is performed afterwards, and the results are shown in Figure 24. The wind retrieval result with all WVCs (Figure 24 (a)) looks quite similar to the wind retrieval result without azimuth averaging (Figure 17 (a)). However, the wind retrieval results get worse when excluding outer WVCs (Figure 24 (b) and Figure 17 (b)) and excluding both outer and nadir WVCs (Figure 24 (c) and Figure 17 (c)). This leads to a conclusion that the averaging of the number of views with similar azimuth angle into a single view is not able to improve the wind retrieval for rotating fan-beam and even if the azimuth diversity of the views is similar, it still adds geometry diversity into the wind retrieval procedure which leads to a better retrieval result.

2) Page 8, line 5: ... are planned to be launched in 2018 and 2019 - please rephrase as plans for a launch in 2018 don't make much sense anymore.

It has been changed to

'CFOSAT has been launched on 29[th] Oct 2018, while WindRad is planned to be launched in 2019.'

3) Discussion on the merits of rotating fan-beam instruments versus pencil-beam instruments - I think it would be good to add the main point from this discussion also to the manuscript.

5  Added in the discussion session:

For the rotating fan-beam and rotating pencil-beam, FoMs give quite clear wind retrieval quality comparison. In general, both rotating systems have the same performance pattern through the cross track as mentioned above, while rotating fan-beam has a better performance for all the FoMs comparing to pencil-beam through all WVCs and the nadir swath tends to spread larger for pencil-beam. The obvious reason is that pencil-beam classified the

10  views as fore- and aft- with HH and VV polarization, which are four views maximum in the sweet swath, while because of the fan shape, it is possible to ensure that the number of views and the diversity of the geometry are high.

4) Figure 9, top left: What are the red and blue points?

The explanation has been added to the captain of Figure 9:

'red points are the average value as a function of wind speed of ScatA; blue points are the average value as a function of wind speed of ScatB'.

20  5) The text to this model validation is also not very clear and could use some proof reading and clarification

It has been updated:

[revised manuscript text omitted]
\int|\boldsymbol{v} - \boldsymbol{v_t}|^2 P_{obs}(\boldsymbol{v}|\boldsymbol{v_t}) \times P_{NWP}(\boldsymbol{v} - \boldsymbol{v_t})d^2v}\right)$ and $RMS_{NWP} = \left(\sqrt{\int\int|\boldsymbol{v} - \boldsymbol{v_t}|^2 P_{NWP}(\boldsymbol{v} - \boldsymbol{v_t})d^2v}\right) = \sqrt{2}\sigma_{NWP}$. VRMS quantifies the wind solution's relative RMS about the true wind with respect to the general prior

5  background uncertainty. If its value is 1, then the wind retrieval failed to provide new and useful information in the wind field, i.e., corresponding to $P_{obs}(\boldsymbol{v}|\boldsymbol{v_t})$ =constant.

On the other hand, AMBI is defined to quantify the ability of the scatterometer and its processing to handle ambiguous solutions without a priori NWP model information. It is a ratio of the wind solution output falling outside the general prior wind field constraint, relative to the output falling inside the prior wind field constraint. The lower the ratio, the better the

10  AMBI FoM, where $P_{NWP,max}$ is the maximum probability of $P_{NWP}(\boldsymbol{v} - \boldsymbol{v_t})$.

$$FoM_{AMBI} = \frac{\int P_{obs}(\boldsymbol{v}|\boldsymbol{v_t}) \times \left(P_{NWP,max} - P_{NWP}(\boldsymbol{v} - \boldsymbol{v_t})\right)d^2v}{\int P_{obs}(\boldsymbol{v}|\boldsymbol{v_t}) \times P_{NWP}(\boldsymbol{v} - \boldsymbol{v_t})d^2v} \qquad (3)$$

BIAS quantifies the systematic vector wind bias, again in the context of the background prior, which is the shift of the average location of the output wind solution away from the location of the prior wind caused by skewness in the output wind solutions (4).

$$FoM_{BIAS} = \int (\boldsymbol{v} - \boldsymbol{v_t}) \cdot P_{obs}(\boldsymbol{v}|\boldsymbol{v_t}) \times P_{NWP}(\boldsymbol{v} - \boldsymbol{
[revised manuscript text omitted]